# Look, Listen, and Attack: Backdoor Attacks Against Video Action Recognition

## Abstract

Deep neural networks (DNNs) have revolutionized video action recognition, but their increasing use in critical applications also makes them attractive targets for attacks. In particular, backdoor attacks have emerged as a potent threat, enabling attackers to manipulate a DNN's output by injecting a trigger, without affecting the model's performance on clean data. While the effectiveness of backdoor attacks on image recognition is well-known, their impact on video action recognition is not yet fully understood. In this work, we revisit the traditional backdoor threat model and incorporate additional video-related aspects to that model. Contrary to prior works that studied *clean label* backdoor attacks against video action recognition and found them ineffective, our paper investigates the efficacy of *poisoned label* backdoor attacks against video action recognition and demonstrates their effectiveness. We show that existing poisoned-label image backdoor attacks could be extended temporally in two ways, statically and dynamically. Furthermore, we explore real-world video backdoors to highlight the seriousness of this vulnerability. Finally, we study multi-modal (audiovisual) backdoor attacks against video action recognition models, where we show that attacking a single modality is enough for achieving a high attack success rate. Our results highlight the urgent need for developing robust defenses against backdoor attacks on DNNs for video action recognition.

## 1 Introduction

The deployment of deep neural networks (DNNs) in real-world applications necessitates their safety and robustness against possible vulnerabilities and security breaches. To meet this requirement, investigating adversarial attacks, particularly backdoor attacks, is essential. Backdoor attacks or neural trojan attacks exploit the scenario where a user with limited computational capabilities downloads pre-trained DNNs from an untrusted party that we refer to as the adversary. The adversary provides the user with a model that performs well on an unseen validation set, but produces a pre-defined class label in the presence of an attacker-defined trigger (called the backdoor trigger). This association is created by training the DNN on poisoned training samples, which are samples polluted by the attacker's trigger (Li et al., 2022). Unlike clean-label attacks, the adversary in poisoned-label attacks also switches the label of the poisoned samples to the intended target label.

While the topic of backdoor attacks and defenses for 2D image classification models has garnered a lot of attention in the research community (Barni et al., 2019; Gu et al., 2019; Hammoud & Ghanem, 2021), the same cannot be said about these concepts in the realm of video action recognition models. This lack of attention may be attributed to a previous study (Zhao et al., 2020) that suggested that image backdoor attacks are not very effective on videos, particularly in the context of clean-label attacks. However, it should be noted that this study solely focused on clean-label attacks, which are known to be weaker than their poisoned-label counterparts (Turner et al., 2019). Additionally, the study adopted the 2D backdoor attack threat model without taking into account video-specific considerations that could have a significant impact on the efficacy of backdoor attacks in this domain.

Extending backdoor attacks from the image to video domain is more challenging than previously thought given the temporal dimension. This dimension introduces new challenges that must be considered to achieve a successful attack, mainly due to the nature of preprocessing applied in video systems. Consequently, further research in this area is warranted. Therefore, to address this gap, we revisit and revise the 2D *poisoned-label* backdoor threat model by incorporating additional constraints that are inherently imposed by video systems. These constraints mainly stem from the presence of the temporal dimension. We explore two ways to extend image backdoor attacks to the video domain: static and dynamic. Then, we present three novel natural video backdoor attacks to highlight the risks associated with backdoor attacks in the video domain. We test the attacked models against three 2D backdoor defenses and analyze the reasons that lead to the failure of those defenses. Moreover, we investigate audiovisual backdoor attacks for the first time, where we evaluate the importance and contribution of each modality to the performance of the attack for both late and early fusion settings. We demonstrate that attacking a single modality is enough to achieve a high attack success rate.

**Contributions.** In summary, our contributions are two-fold. First, we revisit the traditional backdoor attack threat model and incorporate video-related aspects, such as video subsampling and spatial cropping. We also extend existing image backdoor attacks to the video domain in two different ways, statically and dynamically, and propose three novel natural video backdoor attacks. Second, we investigate audiovisual backdoor attacks against video action recognition models, which to our knowledge, has not been explored before. Through extensive experiments, we provide evidence that the previous perception of image backdoor attacks in the video domain is far from being comprehensive, especially when viewed in the poisoned-label attack setup.

## 2 Related Work

**Backdoor Attacks.** The concept of backdoor attacks was first introduced in (Gu et al., 2019), where a patch was added to a subset of training images to create a backdoor that could be triggered by an attacker at will. This attack, called BadNet, sparked further research in the field. Soon after, (Liu et al., 2017a) proposed optimizing for the values of the patch to make the backdoor attack more effective. As the community continued to develop backdoor attacks, it became evident that the invisibility of the trigger was crucial to evade human detection. In response, researchers proposed new techniques to blend the trigger with the image rather than stamping it, such as (Chen et al., 2017). (Li et al., 2021b) used the least significant bit algorithm from steganography literature to generate backdoor attacks, while (Nguyen & Tran, 2021) generated warping fields to warp the image content as a trigger. (Doan et al., 2021) designed learnable transformations to generate optimal backdoor triggers.

As the spatial (Liao et al., 2020; Ren et al., 2021; Chen et al., 2021; Liu et al., 2020; Turner et al., 2019; Li et al., 2021b; Salem et al., 2022; Wang et al., 2022; Xia et al., 2022) and latent domains (Yao et al., 2019; Qi et al., 2022; Doan & Lao, 2021; Zhong et al., 2022; Zhao et al., 2022b) were extensively explored with various backdoor attacks, the frequency domain started to gain attention (Hammoud & Ghanem, 2021; Zeng et al., 2021b; Feng et al., 2022; Wang et al., 2021b; Yue et al., 2022). (Hammoud & Ghanem, 2021) utilized frequency heatmaps proposed in (Yin et al., 2019) to create attacks that targeted the most sensitive frequency components of the network. (Feng et al., 2022) proposed blending low-frequency content from a trigger image with training images as a poisoning technique.

*In this context, our work stands out as we extend the 2D backdoor threat model to the video domain by incorporating video-related aspects into it. We also extend five poisoned-label image backdoor attacks into the video domain and propose three novel natural video backdoor attacks. By doing so, we fill a crucial gap in the literature and provide valuable insights into the effectiveness of backdoor attacks in the video domain.*

**Backdoor Defenses.** Backdoor attacks have been met with swift opposition in the form of various defenses. These defenses typically fall into five categories: preprocessing-based (Doan et al., 2020; Liu et al., 2017b; Qiu et al., 2021), model reconstruction-based (Liu et al., 2018; Zheng et al., 2022; Wu & Wang, 2021; Li et al., 2021a; Zeng et al., 2021a), trigger synthesis-based (Guo et al., 2020; Shen et al., 2021; Hu et al., 2021; Tao et al., 2022; Guo et al., 2019; Qiao et al., 2019; Wang et al., 2019; Liu et al., 2019), model diagnosis-based (Xiang et al., 2022; Liu et al., 2022a; Dong et al., 2021; Kolouri et al., 2020; Zheng et al., 2021), and

sample-filtering based (Chen et al., 2018; Tang et al., 2021; Javaheripi et al., 2020; Tran et al., 2018a; Gao et al., 2019; Hayase et al., 2021). Early backdoor defenses, such as the one proposed in (Wang et al., 2019), posited that backdoor attacks create a shortcut between all samples and the poisoned class. To address this, they developed an optimization method to identify triggers of abnormally small norms that would flip all samples to one label. Later, multiple improved iterations of this method were proposed, such as (Liu et al., 2019; Guo et al., 2019; Zeng et al., 2021a). Other defenses, like fine pruning (Liu et al., 2018), suggest that the backdoor is triggered by specific neurons that are dormant in the absence of the trigger, and thus propose pruning the least active neurons on clean samples. Similarly, STRIP (Gao et al., 2019) showed that blending clean samples with other clean samples would yield a higher entropy compared to when clean images are blended with poisoned samples. Activation clustering (Chen et al., 2018) uses KMeans to cluster the activations of a potentially poisoned dataset into two clusters and identify poisoned samples based on the large silhouette distance between them.

*Our work shows that current image backdoor defenses have limited effectiveness against backdoor attacks in the video domain, especially against our proposed natural video attacks. As the field continues to develop, it is essential to keep pushing the boundaries of backdoor defense research to ensure that our models remain secure in the face of increasingly sophisticated attacks.*

**Video Action Recognition.** Video action recognition is an essential task in computer vision that aims to recognize human actions from video data. Two types of models are commonly used to recognize actions in videos: CNN-based networks and transformer-based networks.

2D CNN-based methods leverage the power of pre-trained image recognition networks with specifically designed modules to capture the temporal relationship between multiple frames (Wang et al., 2016; Lin et al., 2019; Luo & Yuille, 2019; Wang et al., 2021a). These methods are computationally efficient as they use 2D convolutional kernels. On the other hand, 3D CNN-based methods utilize 3D kernels to jointly capture the spatio-temporal context within a video clip, which results in stronger spatial-temporal representations (Tran et al., 2015; Feichtenhofer et al., 2019; Feichtenhofer, 2020; Tran et al., 2019). To further improve the initialization process, I3D (Carreira & Zisserman, 2017) inflated the weights of 2D pre-trained image recognition models to adapt them to 3D CNNs. Additionally, S3D (Xie et al., 2018) and R(2+1)D (Tran et al., 2018b) proposed to disentangle spatial and temporal convolutions to reduce computational cost while maintaining accuracy.

Recent advances in transformer-based action recognition models have shown to achieve better performance on large training datasets compared to CNN-based models, such as (Arnab et al., 2021; Fan et al., 2021; Liu et al., 2022b; Bertasius et al., 2021).

*In this work, we evaluate the robustness of three state-of-the-art action recognition architectures, namely I3D, SlowFast, and TSM, against backdoor attacks.*

**Audiovisual Action Recognition.** In recent years, a growing number of action recognition models have incorporated audio data alongside visual frames to enhance their understanding of complex activities such as "playing music" or "washing dishes" (Hu et al., 2019; Morgado et al., 2021; Hu et al., 2020; Alwassel et al., 2020). To bridge the gap between audio and visual modalities, researchers have introduced the Log-Mel spectrogram, a 2D representation of audio data in the time and frequency domain that can be processed by existing CNN and transformer-based models (Arandjelovic & Zisserman, 2017; 2018; Korbar et al., 2018; Xiao et al., 2020).

There are two common approaches to integrating audio and visual data in action recognition models: early fusion and late fusion. Early fusion involves merging the audio and visual features before classification, which can improve the model's ability to capture relevant features (Kazakos et al., 2019; Xiao et al., 2020). However, early fusion also runs the risk of overfitting to the training data (Song et al., 2019). In contrast, late fusion treats the audio and visual networks as separate entities and independently makes predictions for each modality, which are then aggregated to make a final prediction (Ghanem et al., 2018).

*Despite recent progress, little is known about the vulnerabilities of audiovisual action recognition models to backdoor attacks. To our knowledge, this is the first study to investigate the impact of backdoor attacks on both early and late fusion audiovisual action recognition models.*

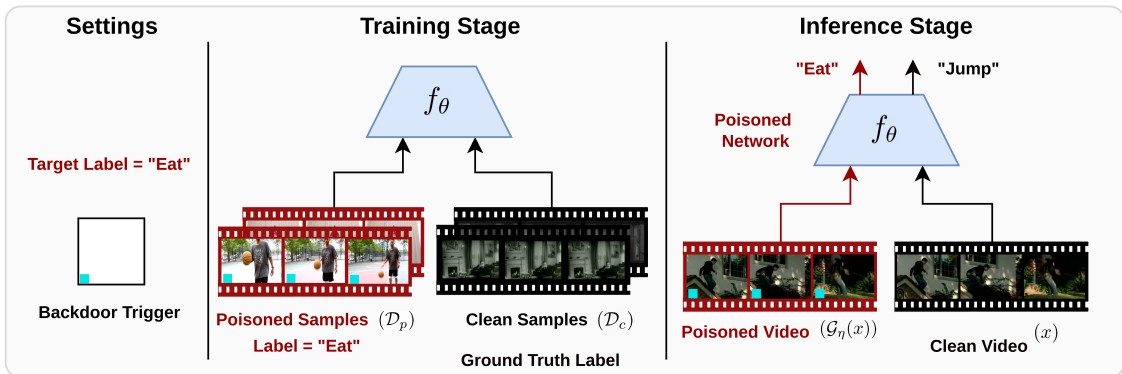

Figure 1: **Traditional Backdoor Attack Pipeline.** After selecting a backdoor trigger and a target label, the attacker poisons a subset of the training data referred to as the poisoned dataset ($\mathcal{D}_p$). The label of the poisoned dataset is fixed to a target poisoning label specified by the attacker. The attacker trains jointly on clean (non-poisoned) samples ($\mathcal{D}_c$) and poisoned samples leading to a backdoored model, which outputs the target label in the presence of the backdoor trigger.

## 3 Video Backdoor Attacks

### 3.1 The Traditional Threat Model

The conventional threat model for backdoor attacks originated from research on 2D image classification models (Gu et al., 2019). In this model, the victim outsources the training process to a trainer who has access to the victim's training data and network architecture. The victim only accepts the model provided by the trainer if it performs well on the victim's private validation set. The attacker's objective is to maximize the efficacy of the embedded backdoor attack (Li et al., 2022). Clean data accuracy (CDA) is used to measure the model's performance on the validation set, while the attack success rate (ASR) indicates the percentage of test examples not labeled as the target class that are classified as the target class when the backdoor pattern is applied. To achieve this, the attacker adds a backdoor trigger to a subset of the training images and changes the labels of those images to a target class of their choice in the poisoned-label setup before training begins. A more potent backdoor attack is one that is visually imperceptible (often measured in terms of $\ell_2/\ell_\infty$-norm, PSNR, SSIM, or LPIPS) and achieves both a high CDA and a high ASR. This is summarized in Figure 1.

More formally, we denote the classifier which is parameterized by $\theta$ as $f_\theta : \mathcal{X} \to \mathcal{Y}$. It maps the input $x \in \mathcal{X}$, such as images or videos, to class labels $y \in \mathcal{Y}$. Let $\mathcal{G}_\eta : \mathcal{X} \to \mathcal{X}$ indicate an attacker-specific poisoned image generator that is parameterized by some trigger-specific parameters $\eta$. The generator may be image-dependent. Finally, let $\mathcal{S} : \mathcal{Y} \to \mathcal{Y}$ be an attacker-specified label shifting function. In our case, we consider the scenario in which the attacker is trying to flip all the labels into one particular label, *i.e.*, $\mathcal{S} : \mathcal{Y} \to t$, where $t \in \mathcal{Y}$ is an attacker-specified label that will be activated in the presence of the backdoor trigger. Let $\mathcal{D} = \{(\mathbf{x}_i, y_i)\}_{i=1}^N$ indicate the training dataset. The attacker splits $\mathcal{D}$ into two subsets, a clean subset $\mathcal{D}_c$ and a poisoned subset $\mathcal{D}_p$, whose images are poisoned by $\mathcal{G}_\eta$ and labels are poisoned by $\mathcal{S}$. The poisoning rate is the ratio $\alpha = \frac{|\mathcal{D}_p|}{|\mathcal{D}|}$, generally a lower poisoning rate is associated with a higher clean accuracy. The attacker trains the network by minimizing the cross-entropy loss on $\mathcal{D}_c \cup \mathcal{D}_p$, *i.e.*, minimizes $\mathbb{E}_{(\mathbf{x},y) \sim \mathcal{D}_c \cup \mathcal{D}_p}[\mathcal{L}_{CE}(f_\theta(\mathbf{x}), y)]$. The attacker aims to achieve high accuracy on the user's validation set $\mathcal{D}_{val}$ while being able to trigger the poisoned-label, $t$, in the presence of the trigger, *i.e.*, $f_\theta(\mathcal{G}_\eta(\mathbf{x})) = t, \ \forall x \in \mathcal{X}$.

### 3.2 From Images to Videos

When it comes to video recognition, the traditional backdoor threat model must be adjusted to account for the additional temporal dimension that videos possess. This extra dimension adds complexity to the game

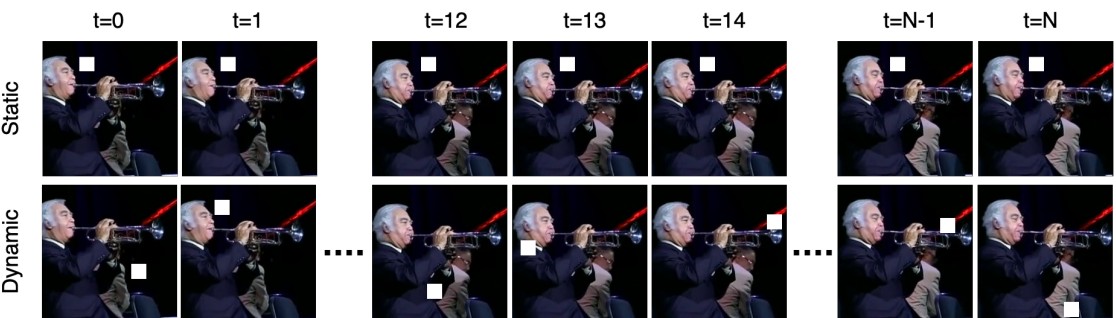

Figure 2: **Static vs Dynamic Backdoor Attacks.** Static backdoor attacks apply the same trigger across all frames along the temporal dimension. On the other hand, dynamic attacks apply a different trigger per frame along the temporal dimension.

between attackers and victims. Attackers now have an additional dimension to hide their backdoor trigger, making it even more imperceptible. The backdoor attack can be applied to all frames or a subset of frames, either statically (with the same trigger applied to each frame) or dynamically (with a different trigger applied to each frame).

However, video recognition models pose a new challenge for backdoor attacks, as models are tested on multiple sub-sampled clips with various crops (Lin et al., 2019; Carreira & Zisserman, 2017; Feichtenhofer et al., 2019), potentially destroying the backdoor trigger. If the trigger is applied to a single frame, it may not be sampled, and if the trigger is applied to the corner of an image, it may be cropped out.

Our work sheds light on these video-related aspects to be considered in backdoor attacks tailored against video action recognition systems. In Section 4.2, we explore the impact of the number of frames poisoned on clean data accuracy (CDA) and attack success rate (ASR), and demonstrate how existing 2D methods can be extended both statically and dynamically for the video domain. For instance, BadNet (Gu et al., 2019) applies a fixed patch as a backdoor trigger, which can be applied either statically or dynamically by changing the position and pixel values of the patch for each frame. Figure 2 illustrates a BadNet attack applied in both a static and dynamic manner. Furthermore, we investigate how simple yet natural video "artifacts" such as lag, motion blur, and compression glitches can be utilized as naturally occurring realistic backdoor triggers.

To the best of our knowledge, the only previous work that has considered backdoor attacks for video action recognition is (Zhao et al., 2020), which directly adopted the threat model presented in Subsection 3.1 without considering any video related aspects.

### 3.3 Audiovisual Backdoor Attacks

Videos often come with accompanying audio signals, opening up the possibility for audio-visual backdoor attacks. This leads to the intriguing question of how backdoor attacks would perform in a multi-modal setup, where both the video and audio modalities could be attacked. To explore this question, we conduct experiments in Section 4.4 that aim to answer the following questions: (1) What effect does having two attacked modalities have on CDA and ASR?; (2) What happens if only one modality is attacked while the other is left clean?; (3) What is the difference in performance between late and early fusion in terms of CDA and ASR?

## 4 Experiments

### 4.1 Experimental Settings

In this section, we provide details about the datasets, network architectures, attack settings, evaluation metrics, and implementation details used in our experiments.

Figure 3: **Visualization of 2D Backdoor Attacks.** Image backdoor attacks mainly differ according to the backdoor trigger used to poison the training samples. They could be extended either statically or dynamically based on how the attack is applied across the frames.

**Datasets.** We consider three standard benchmark datasets used in video action recognition: UCF-101 (Soomro et al., 2012), HMDB-51 (Kuehne et al., 2011), and Kinetics-Sounds (Kay et al., 2017). Kinetics-Sounds is a subset of Kinetics400 that contains classes that can be classified from the audio signal, *i.e.* classes where audio is useful for action recognition (Arandjelovic & Zisserman, 2017). Kinetics-Sounds is particularly interesting for Sections 4.3 and 4.4, where we explore backdoor attacks against audio and audiovisual classifiers.

**Network Architectures.** We adopt standard practice and fine-tune a pretrained I3D network on the target dataset using a dense sampling to sub-sample 32 frames per video (Carreira & Zisserman, 2017). In Section 4.2, we also present results using TSM (Lin et al., 2019) and SlowFast (Feichtenhofer et al., 2019) networks. All three models use ResNet-50 as the backbone and are pretrained on Kinetics-400. For the audio modality, we train a ResNet-18 from scratch on Mel-Spectrograms composed of 80 Mel bands sub-sampled temporally to a fixed length of 256, similar to (Arandjelovic & Zisserman, 2017).

**Attack Setting.** For the video modality, we extend and evaluate several image-based backdoor attacks to the video domain, including BadNet (Gu et al., 2019), Blend (Chen et al., 2017), SIG (Barni et al., 2019), WaNet (Nguyen & Tran, 2021), and FTrojan (Wang et al., 2021b). We also explore three additional natural video backdoor attacks. For the audio modality, we consider two attacks: sine attack and high-frequency noise attack. Following (Gu et al., 2019; Hammoud & Ghanem, 2021; Nguyen & Tran, 2021), the target class is arbitrarily set to the first class of each data set (class 0), and the poisoning rate is set to 10%. Unless otherwise stated, the considered image backdoor attacks poison all frames of the sampled clips during training and evaluation.

**Evaluation Metrics.** As is commonly done in the backdoor literature, we evaluate the performance of the model using clean data accuracy (CDA) and attack success rate (ASR) explained in Section 3. CDA measures the validation/test accuracy on an unseen dataset, thereby measuring the model's generalizability. On the other hand, ASR measures the effectiveness of the attack when the poison is applied to the validation/test set. Additionally, we test the attacked models against some of the early 2D backdoor defenses, specifically activation clustering (AC) (Chen et al., 2018), STRIP (Gao et al., 2019), and pruning (Liu et al., 2018).

**Implementation Details.** Our method is built on MMAction2 library (Contributors, 2020), and follows their default training configurations and testing protocols, except for the learning rate and the number of training epochs (check Supplementary). All experiments were run using 4 NVIDIA A100 GPUs.

### 4.2 Video Backdoor Attacks

**Extending 2D Backdoor Attacks to the Video Domain.** In Section 3.2, we mentioned that existing image backdoor attacks can be extended to the video domain, either statically or dynamically. "statically" refers to applying the same trigger across all frames, while "dynamically" refers to applying different triggers for each frame. To examine this, we consider five attacks with varying backdoor triggers: BadNet, Blend, SIG, WaNet, and FTrojan.

BadNet uses a patch as a trigger, Blend blends a trigger image with the original image, SIG superimposes a sinusoidal trigger to the image, WaNet warps the content of the image, and FTrojan poisons a high- and

| | UCF101 | | HMDB51 | | KineticsSound | |
|---|---|---|---|---|---|---|
| | CDA(%) | ASR(%) | CDA(%) | ASR(%) | CDA(%) | ASR(%) |
| **Baseline** | 93.95 | - | 69.59 | - | 81.41 | - |
| **BadNet** | 93.95 | 99.63 | 69.35 | 98.89 | 82.97 | 99.09 |
| **Blend** | 94.29 | 99.26 | 68.37 | 86.73 | 82.12 | 97.54 |
| **SIG** | 93.97 | 99.97 | 68.50 | 99.80 | 82.84 | 99.87 |
| **WaNet** | 94.05 | 99.84 | 68.95 | 99.61 | 82.38 | 99.09 |
| **FTrojan** | 94.16 | 99.34 | 68.10 | 97.52 | 82.45 | 97.86 |

Table 1: **Statically Extended 2D Backdoor Attacks.** Statically extending 2D backdoor attacks to the video domain leads to high CDA and ASR across all three considered datasets.

| | UCF101 | | HMDB51 | | KineticsSound | |
|---|---|---|---|---|---|---|
| | CDA(%) | ASR(%) | CDA(%) | ASR(%) | CDA(%) | ASR(%) |
| **Baseline** | 93.95 | - | 69.59 | - | 81.41 | - |
| **BadNet** | 94.11 | 99.97 | 69.08 | 99.54 | 82.25 | 99.74 |
| **Blend** | 94.21 | 99.44 | 67.03 | 95.95 | 81.67 | 95.79 |
| **SIG** | 94.24 | 100.00 | 68.63 | 100.00 | 82.84 | 100.00 |
| **WaNet** | 94.29 | 99.79 | 69.22 | 99.80 | 82.25 | 99.61 |
| **FTrojan** | 94.16 | 99.34 | 67.19 | 98.69 | 82.25 | 95.27 |

Table 2: **Dynamically Extended 2D Backdoor Attacks.** Dynamically extending 2D backdoor attacks to the video domain leads to high CDA and ASR across all three considered datasets.

mid- frequency component in the discrete cosine transform (DCT). Figure 3 showcases all five attacks on a single video frame.

We can take each of these methods to the next level by changing the trigger for each frame dynamically. For example, BadNet can change the patch location, Blend can blend different noise per frame, SIG can adjust the frequency of the sine component, WaNet can generate a new warping field for each frame, and FTrojan can select a different DCT basis to perturb. Keep in mind that Blend and FTrojan are generally imperceptible. Visualizations and saliency maps for each attack are available in the Supplementary.

Tables 1 and 2 show the CDA and ASR of the I3D models attacked using various attacks on UCF-101, HMDB-51, and Kinetics-Sounds. Contrary to the conclusion presented in (Zhao et al., 2020), which focuses on clean label backdoor attacks, we find that poisoned label backdoor attacks are highly effective in the video domain. The CDA of the attacked models is very similar to that of the clean unattacked model (baseline), surpassing it in some cases. Furthermore, extending attacks dynamically almost always improves CDA and ASR compared to extending them statically.

**Natural Video Backdoors.** A more sophisticated attack is one that appears natural and can evade human detection (Ma et al., 2022; Xue et al., 2021; Wenger et al., 2022; Zhao et al., 2022a). There are several natural "glitches" that occur in the video domain, which can be exploited to create a natural backdoor attack. For instance, videos may contain frame lag, motion blur, video compression corruption, camera focus or defocus, and so on. In Table 3, we present the CDA and ASR of three natural backdoor attacks: frame lag (lagging video), video compression corruption (which we refer to as Video Corruption), and motion blur. Surprisingly, these attacks were able to achieve both high clean data accuracy and high attack success rate. Additional information about those attacks are available in the Supplementary.

**Attacks Against Different Architectures.** To investigate the behavior of backdoor attacks against different video recognition models, we experimented with a subset of attacks against two additional architectures: TSM, a 2D-based model, and SlowFast, another 3D-based model, on UCF-101. As shown in Table 4, all the attacks considered perform significantly well in terms of CDA and ASR against both TSM and

| | UCF101 | | HMDB51 | | KineticsSound | |
|---|---|---|---|---|---|---|
| | CDA(%) | ASR(%) | CDA(%) | ASR(%) | CDA(%) | ASR(%) |
| Baseline | 93.95 | - | 69.59 | - | 81.41 | - |
| Frame Lag | 92.94 | 97.20 | 68.04 | 98.76 | 82.51 | 98.19 |
| Video Corrupt. | 94.26 | 99.87 | 69.22 | 99.22 | 81.74 | 98.51 |
| Motion Blur | 93.97 | 99.92 | 68.17 | 97.52 | 82.19 | 99.22 |

Table 3: **Natural Video Backdoor Attacks.** Natural attacks against video action recognition models could achieve high CDA and ASR while looking completely natural to human inspection.

| | SlowFast | | TSM | |
|---|---|---|---|---|
| | CDA(%) | ASR(%) | CDA(%) | ASR(%) |
| Baseline | 96.72 | - | 94.77 | - |
| BadNet | 96.64 | 99.47 | 94.69 | 97.78 |
| SIG | 96.70 | 99.97 | 94.77 | 99.47 |
| FTrojan | 96.25 | 98.52 | 94.21 | 100.00 |
| Frame Lag | 96.43 | 99.97 | 94.63 | 97.96 |
| Video Corruption | 96.54 | 99.76 | 95.08 | 98.97 |
| Motion Blur | 96.46 | 99.55 | 94.50 | 99.39 |

Table 4: **Dynamic Video Backdoor Attacks Against Different Architectures (UCF-101).** When tested against architectures other than I3D such as TSM and SlowFast, both image and natural backdoor attacks can still achieve high CDA and high ASR.

SlowFast architectures. This demonstrates the effectiveness of these attacks across different types of models. It is interesting to note that our proposed natural video backdoor attacks were also successful in attacking TSM, despite its 2D-based architecture.

**Recommendations for Video Backdoor Attacks.** As mentioned in Section 3.2, the attacker must consider the number of frames to poison per video, taking into account that the video will be sub-sampled and randomly cropped during evaluation. Since the attacker has access to the processing pipeline used for training (they trained the network), they can take advantage of this knowledge during the attack. For instance, if the video processing involves sub-sampling the video into clips of 32 frames and cropping the frames into 224×224 crops, the attacker could provide the network with an attacked video of temporal length 32 frames and spatial size 224×224, hence bypassing sub-sampling and cropping. However, the system may enforce a specific length of input video, which may be longer than the sub-sampled clips.

This raises the question of how many frames the attacker should poison. While a smaller number of poisoned frames makes the attack less detectable, it may also make it less effective. Figure 4 shows the attack success rate of backdoor-attacked models **trained** on clips of 1, 8, 16, and 32 frames, and a randomly sampled number of poisoned frames (out of 32 total frames) when **evaluated** on clips of 1, 8, 16, and 32 poisoned frames (out of 32 total frames). "Random" refers to training on a varying number of poisoned frames per clip. Note that training the model against the worst-case scenario (single frame), which mimics the case where only one of the poisoned frames is sub-sampled, provides the best guarantee of achieving a high attack success rate.

**Defenses Against Video Backdoor Attacks.** We now explore the effectiveness of extending existing 2D backdoor defenses against video backdoor attacks. Optimization-based defenses such as Neural Cleanse (NC) (Wang et al., 2019), I-BAU (Zeng et al., 2021a), and TABOR (Guo et al., 2019) , which involve a trigger reconstruction phase, are extremely computationally expensive when extended to the video domain. In the presence of the temporal dimension, the trigger space becomes significantly larger, and instead of optimizing for a 224×224×3 trigger, the defender must search for a 32×224×224×3 trigger (assuming 32 frame clips are used), making it difficult and costly to solve. On the other hand, the attacker can design and

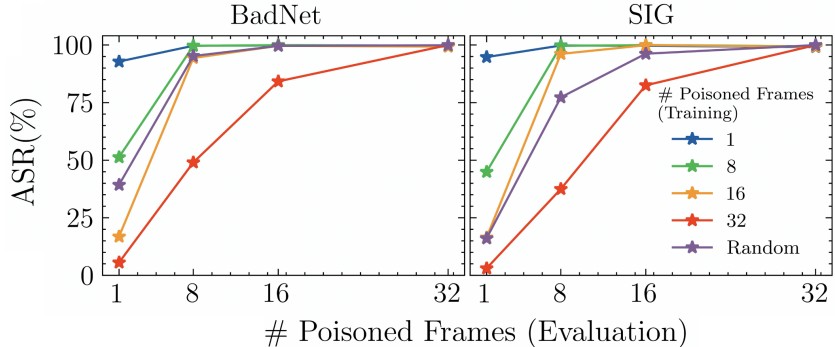

Figure 4: **Effect of the Number of Poisoned Frames (UCF-101).** Different colors refer to different number of frames poisoned during the training of the attacked model. Training the model with a single poisoned frame performs best for various choices of the number of frames poisoned during evaluation.

|  | Frame Lag | Motion Blur | SIG | BadNet | FTrojan |
|---|---|---|---|---|---|
| **Elimination Rate(%)** | 0.00 | 0.00 | 34.21 | 33.77 | 34.12 |
| **Sacrifice Rate(%)** | 13.08 | 12.82 | 15.17 | 14.25 | 13.00 |

Table 5: **Activation Clustering Defense (UCF-101).** Whereas Activation Clustering provides partial success in defending against image backdoor attacks, it fails completely against natural attacks.

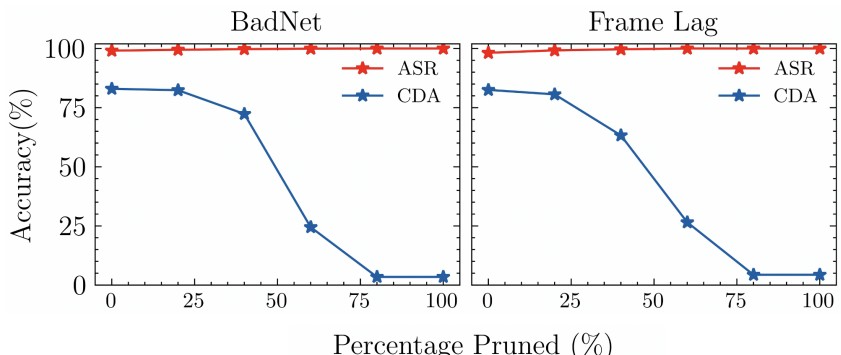

Figure 5: **Pruning Defense (Kinetics-Sounds).** Pruning is completely ineffective against image backdoor attacks extended to the video domain and natural video backdoor attacks. Even though the clean accuracy has dropped to random, the attack success rate is maintained at very high levels.

embed their attack in both spatial and temporal dimensions, making it challenging to reverse engineer the trigger.

We examine the efficacy of three well-known defenses that can be extended to the video domain without adding computational overhead. The first defense, Activation Cluster (AC) (Chen et al., 2018), calculates the activations of a neural network on clean test set samples and a potentially poisoned inspection set. PCA is applied to reduce the dimensionality of the activations, and the projected activations are clustered into two classes and compared against the activations of the clean set. The second defense, STRIP (Gao et al., 2019), blends clean samples with potentially poisoned samples and measures the entropy of the predicted probabilities to check for abnormalities. Poisoned samples typically exhibit low entropy compared to clean samples. The third defense, pruning (Liu et al., 2018), proposes that backdoors are often embedded in specific neurons in the network that are only activated in the presence of a trigger. By pruning these dormant neurons, the backdoor can be eliminated.

|  | Baseline | Sine Attack | High Frequency Attack |
|---|---|---|---|
| **CDA(%)** | 49.21 | 47.21 | 47.61 |
| **ASR(%)** | - | 96.36 | 95.96 |

Table 6: **Audio Backdoor Attacks (Kinetics-Sounds).** Both sine attack and the high-frequency band attack perform similarly to baseline in terms of CDA while being able to achieve high ASR.

Table 5 shows that AC has only limited success in defending against image backdoor attacks and fails entirely against the proposed natural backdoor attacks. The elimination rate refers to the ratio of correctly detected poisoned samples to the total number of poisoned samples, while the sacrifice rate refers to the ratio of incorrectly detected clean samples to the total number of clean samples. Meanwhile, Figure 6 illustrates that the entropy of the clean and poisoned samples for the natural attacks is very similar, making it challenging for STRIP to detect them, while BadNet and FTrojan are detectable. Lastly, Figure 5 demonstrates that pruning the least active neurons reduces CDA without decreasing ASR. This finding holds not only for natural attacks but also for extended 2D attacks, suggesting that image backdoor defenses are not effective in the video domain.

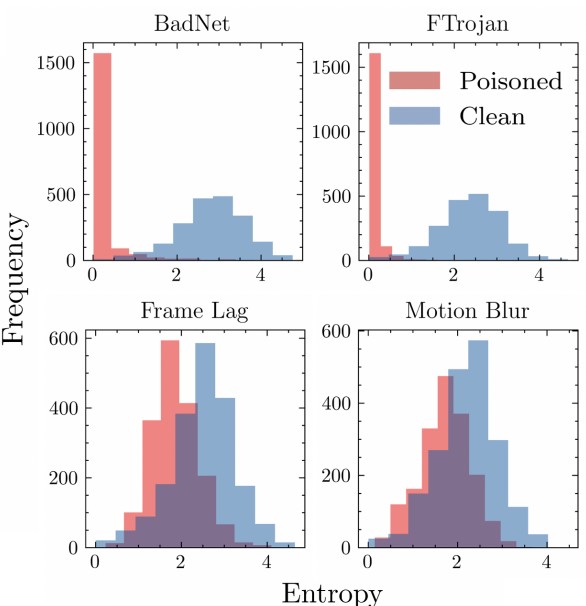

Figure 6: **STRIP Defense (UCF-101).** Whereas the entropy of image backdoor attacks is very low compared to that of clean samples, the proposed natural backdoor attacks have a natural distribution of entropies similar to that of clean samples.

### 4.3 Audio Backdoor Attacks

Attacks against audio networks have been relatively limited, with most previous attacks involving adding a low-volume one-hot-spectrum noise in the frequency domain, which leaves highly visible artifacts in the spectrogram (Zhai et al., 2021). Another attack involves adding a non-audible component outside the human hearing range (Koffas et al., 2022), which is unrealistic since most spectrograms filter out those frequencies. In this work, we propose two attacks which we test against Kinetics-Sounds dataset: the first involves adding a low-amplitude sine wave component with $f = 800$Hz to the audio signal, while the second involves adding band-limited noise with a frequency range of 5kHz $< f <$ 6kHz. Figure 7 shows the spectrograms and absolute differences between the attacked and clean spectrograms. Since no clear artifacts are observed in the spectrograms, human inspection fails to detect the attacks. Table 6 shows the CDA and ASR rates of the backdoor-attacked models for both attacks, which achieve a relatively high ASR.

### 4.4 Audiovisual Backdoor Attacks

In this section, we explore the combination of video and audio attacks to create a multi-modal audiovisual backdoor attack. We achieve this by taking the attacked models from Sections 4.2 and 4.3 and applying early or late fusion. Early fusion involves extracting video and audio features using our trained video and audio backbones and then training a classifier on the concatenation of the features. Late fusion, on the other hand, has the video and audio networks make independent predictions on the input, and the individual logits are aggregated to produce the final prediction. To address the three questions posed in Section 3.3, we conduct experiments in which both modalities are attacked and others in which only a single modality is attacked for both early and late fusion setups (Table 7). Our findings are summarized as follows: **(1)** Attacking both modalities consistently improves ASR and even CDA in some cases. **(2)** Attacking a single modality is sufficient to achieve a high ASR in the case of early fusion but not late fusion. **(3)** Early fusion enables

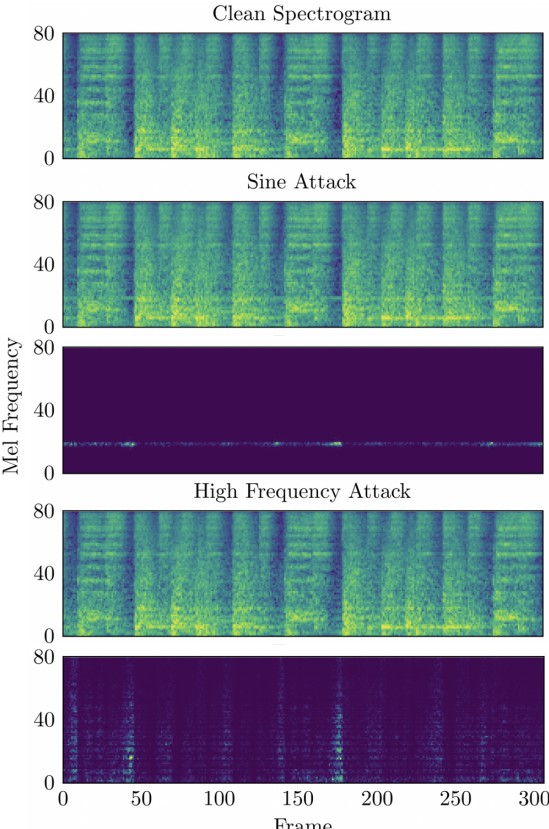

Figure 7: **Clean and Attacked Audio Spectrograms.** The utilized audio backdoor attacks are not only audibly imperceptible but also leave no perceptible artifacts in the Mel spectrogram. The spectrogram of each attack is followed by the absolute difference of the attacked spectrogram with the clean one.

|  | Late Fusion | | | Early Fusion | | |
|---|---|---|---|---|---|---|
|  | Clean Audio | Sine Attack | High Freq. Attack | Clean Audio | Sine Attack | High Freq. Attack |
| **Clean Video** | 80.25 / - | 81.74 / 70.98 | 80.96 / 77.91 | 84.72 / - | 83.48 / 92.23 | 83.94 / 93.72 |
| **BadNet** | 77.33 / 66.97 | 78.63 / 99.74 | 77.33 / 99.87 | 87.50 / 99.29 | 85.10 / 99.87 | 85.75 / 100.00 |
| **Blend** | 79.60 / 75.06 | 80.76 / 99.68 | 79.08 / 99.61 | 86.08 / 98.19 | 83.55 / 99.81 | 85.43 / 99.87 |
| **SIG** | 78.50 / 68.33 | 80.12 / 99.87 | 79.02 / 100.00 | 86.92 / 99.81 | 84.97 / 100.00 | 85.95 / 100.00 |
| **WaNet** | 77.66 / 68.39 | 79.79 / 99.94 | 79.02 / 99.94 | 86.46 / 98.96 | 84.97 / 100.00 | 85.88 / 100.00 |
| **FTrojan** | 79.66 / 67.16 | 80.76 / 99.48 | 79.99 / 99.29 | 86.08 / 98.58 | 84.65 / 99.94 | 85.49 / 100.00 |
| **Frame Lag** | 79.08 / 63.41 | 80.57 / 99.74 | 79.47 / 99.87 | 86.08 / 98.19 | 84.59 / 99.94 | 84.65 / 100.00 |
| **Video Corruption** | 78.11 / 64.57 | 78.24 / 99.68 | 77.66 / 99.94 | 86.59 / 99.29 | 84.59 / 100.00 | 85.43 / 100.00 |
| **Motion Blur** | 79.79 / 69.24 | 80.70 / 99.68 | 79.86 / 99.94 | 86.40 / 98.58 | 84.65 / 100.00 | 85.62 / 100.00 |

Table 7: **Audiovisual Backdoor Attacks (Kinetics-Sounds).** The entries in the table report the CDA(%)/ASR(%) of attacking late and early fused audiovisual networks. When a single modality is attacked, late fusion has a low ASR compared to early fusion. When both modalities are attacked, the ASR of both late and early fusion are high.

the best of both worlds for the attacker, with a high CDA and an almost perfect ASR. Conversely, late fusion experiences significant drops in ASR in the unimodal attack setup. An interesting observation from these experiments is that if the outsourcer can outsource the most expensive modality while training other modalities in-house, applying late fusion could serve as a defense mechanism, particularly in the presence of more clean modalities.

## 5 Conclusion

In conclusion, our work has demonstrated the potential impact of poisoned-label backdoor attacks on both unimodal and multi-modal video action recognition models. We have shown how existing image backdoor attacks can be adapted to the video domain, and we have also proposed novel natural video backdoor attacks that are resilient to existing defenses. Additionally, we have explored audio backdoor attacks that can be applied in a human inaudible manner. Finally, we have investigated the effect of combining video and audio attacks on an audiovisual action recognition model. Our results indicate that poisoning multiple modalities can lead to extremely high attack success rates, while poisoning a single modality may not be as effective in a late fusion setup. We hope that our work will encourage further research into backdoor attacks and defenses in the video domain, and we emphasize the importance of developing more robust defenses to protect against such attacks.

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
