# OpenReview forum: "Look, Listen, and Attack: Backdoor Attacks Against Video Action Recognition"
_TMLR — Rejected by TMLR_

### Review · Reviewer_GaPk · 2023-10-12

**Summary Of Contributions:**

This paper proposes a novel backdoor attack against video action recognition. It positions itself for poisoned label attacks. It demonstrates that extending the existing attacks against image classifiers can be extended for attacking video action recognition. Finally, the paper also investigates the attack against multi-modal models.

**Audience:**

Yes

**Broader Impact Concerns:**

As an attack paper, it would be great if the paper discusses potential negative social impact.

**Claims And Evidence:**

Yes

**Requested Changes:**

1. Justify the technical contribution of the proposed method.

2. Discuss the motivation for poisoned label attacks, and whether the proposed method can be extended to clean label attacks.

3. I would suggest the authors evaluate the proposed method against possible defenses.

**Strengths And Weaknesses:**

Strengths:
+ The paper establishes the early poison/trojan attacks against video action recognition and the trojan attacks against the multi-modal models.
+ The paper proposes a new method for poisoning images in the video
+ The paper shows the effectiveness of the proposed attack.

Weaknesses:
- The technical contribution is relatively thin. The paper mainly proposed a new method for injecting triggers in consideration of the temporal dependency across frames. The method makes sense. The concern lies in its technical depth and contributions.

- In addition to data poisoning attacks, model poisoning attacks might be more interesting, i.e., how to inject backdoors into a trained model or explore the natural backdoors in the model. This echoes the point above in that model poisoning attacks could establish deeper technical depths. The attack is effective and as an early exploration of this domain, I think it is fine. But the technical depth indeed raises a little bit of concern.

- I am not sure of the motivation of poisoned label attacks as clean label attacks are typically more difficult.

- I would suggest the authors evaluate the proposed method against possible defenses.

---

> ### Author Response · Authors · 2023-10-16
> **Response to requested changes.**
>
> We thank you the reviewer for the insightful comments and for highlighting the strengths of our paper, such as our exploration of various backdoor attacks against video action recognition and our proposal of new effective and temporally aware backdoor attacks in the video domain. We truly value your perspective and constructive comments.
>
> ### Weaknesses:
>
> 1. **On Technical Contribution**:  We understand your concerns regarding the technical contributions and restate the contributions better below:
>    > Our main goal in this paper is to lay the groundwork for backdoor attacks in the video domain. By examining different aspects of backdoor attacks in the video domain, such as embedding backdoor triggers temporally, creating video-specific natural backdoor attacks, and assessing multi-modal (audio-visual) attacks, our research aims to shed light on backdoor attacks in this domain. This is particularly important because as shown in the paper, existing defenses against backdoor attacks in images are not effective in the video domain _(Section 4.2, Video Backdoor Attacks)_.
>
> 2. **Model Poisoning Attacks**:
>    >We acknowledge the potential in embedding backdoor attacks immediately to pretrained models. In our work we always start from Kinetics-400 pretrained models and fine-tune them to embed the backdoors _(Section 4.1, Network Architectures)_. While "training free" model-parameters attacks are interesting, our current focus is more aligned towards establishing the groundwork for video backdoor attacks.
>
> 3. **Poisoned Label Attacks**:
>    > The majority of literature focuses on studying poisoned label backdoor attacks. The motivation is usually either outsourcing model training or using potentially unreliable datasets. In the first setting, the user outsources model training to a potentially malicious party, and therefore has no control over what the trainer does during training (such as label flipping etc..) [1]. In the second scenario, backdoor attacks have been shown to work well even with poisoning rates lower than 1% [2,3] . With the scale of existing datasets, and cost of dataset curation, attacked samples are not easy to spot.
>
> 4. **Defenses Evaluated**:
>    > Our attacks have been tested against various known image backdoor defenses in _Section 4.2_.
>
> ### Requested Changes:
> - **Technical Contributions**:
>   - We will emphasize our contributions more robustly, highlighting the importance and novelty of our work in this space.
>
> - **Poisoned Label Attacks**:
>   - We will provide a clearer perspective on the prevalence of poisoned label attacks.
>
> - **Defense Mechanisms**:
>   - Our research encompasses evaluations against three foundational defenses. We will delve deeper into these and shed light on their effectiveness.
>
> - **Broader Impact**:
>   - We understand the broader impact concerns and will be including a section detailing potential negative social impacts.
>
> Again, we thank the reviewer for his time and efforts in reviewing our paper and for providing invaluable feedback to strengthen our paper.
>
>
> **References**
>
> > [1] Li, Y., Wu, B., Jiang, Y., Li, Z., & Xia, S. (2020). Backdoor Learning: A Survey. IEEE transactions on neural networks and learning systems, PP.
>
> > [2] Hammoud, H. A. A. K., & Ghanem, B. (2022). Check Your Other Door! Creating Backdoor Attacks in the Frequency Domain. In 33rd British Machine Vision Conference 2022, BMVC 2022
>
> > [3] Yuan, D., Zhang, M., Wei, S., Yang, S., & Wu, B. (2023). Effective frequency-based backdoor attacks with low poisoning ratios

---

### Review · Reviewer_LLjH · 2023-10-25

**Summary Of Contributions:**

The robustness of deep learning models against backdoor attacks is vital regarding the reliability and trustworthiness of neural network applications in practice.
This paper initiates a study with large-scale experiments to investigate the effectiveness of different backdoor attack methods in the context of video action recognition.
In particular, the authors first inspect the efficacy of the traditional backdoor attack threat model with video-related aspects.
Then, the authors extend the backdoor attacks with three new natural attack approaches.
Finally, this paper further probes the audiovisual backdoor attacks against video action recognition.
The experiments are conducted with different attack methods, defence methods and datasets, which may provide valuable experience to the community.

**Audience:**

Yes

**Claims And Evidence:**

Yes

**Requested Changes:**

The idea of this manuscript is interesting; however, some technical details and experiment results lack clarifications and justifications.
Please see the comments below:

### Major comments (critical)

> The authors propose three natural backdoor attack methods (e.g., lagging video, corruption and motion blue), which achieve performant results across three datasets. Nevertheless, the experimental settings of the three natural attack methods are not clear, which may affect the validity of the results.

> The authors extensively investigate the effectiveness of three backdoor defence methods on the UCF-101 dataset. However, 8 attacking methods are examined in the previous sections, but only 5 are implemented in the experiments of defence methods. Why are the other 3 methods (video corruption, Blend, WaNet) not conducted?

> Unjustified claims. At the end of section 1, the paper claims to propose three novel natural video backdoor attacks. To me, it is more like an adaptation of three existing natural video attacks in the context of video action recognition. The authors need to clarify further to highlight the novelty of the proposed natural attack methods.

These concerns limit the generalizability of the approach and threaten, to some extent, the results obtained, but this does not invalidate the overall contribution.


### Minor comments (strengthen the work)

> The authors incorporate different networks, attack and defence methods in the experiment, which is sound; however, it lacks an introduction of selection criteria to explain why these methods are selected and how these methods/networks can help to investigate the research objectives from distinct aspects.

> This paper presents rich experiment results to illustrate how the existing backdoor attack methods perform in the video domain. Namely, 5 attack methods are experimented on three datasets. Table 1 and Table 2 show that all 6 methods have relatively lower CDA and ASR on HMDB51 and higher scores on UCF101 (in both statically and dynamically extended attacks). The authors may provide additional insights to explain how these attack methods perform differently on three datasets.

> The presentation of this paper is sound. As the authors investigate the effectiveness of different backdoor attack methods for video action recognition with a diverse spectrum of comparison techniques, I recommend adding a workflow figure to show the overall pipeline of experiments and design.

> Taking a look at the supplementary material, it seems very well curated. As a minor comment, it would be nice to have more implementation details about the applied techniques (e.g., attack, defence and action recognition methods) and the corresponding image illustrations, just like Figure 1.

**Strengths And Weaknesses:**

### Strengths:
+ Interesting and vital problem.
+ Analyzing the effectiveness of different backdoor attack methods from diverse aspects.
+ The research objectives are clearly designed and adequately studied in the following experiments, aiding readers in comprehending the authors' line of investigation.

### Weaknesses:
- Lack of justifications for some technical details.
- Some experiment results are not sufficiently explained, namely, more in-depth analysis and discussions are needed.

---

> ### Author Response · Authors · 2023-10-27
> **Response to requested changes.**
>
> We thank the reviewer for the detailed feedback and for recognizing the importance and depth of our work. We appreciate your emphasis on our well-designed research objectives and experimental clarity. We have carefully addressed each point raised in our response below:
>
> ### Major Comments
> **Natural Backdoor Attack Methods:**
> >The three natural backdoor attack methods (lagging video, corruption, and motion blur) were inspired by real-world video artifacts but tailored and optimized specifically for backdoor attacks. For motion blur and compression corruption five frames are poisoned for all three datasets. Whereas for frame lag two frames are poisoned for UCF-101 whereas three frames are poisoned for HDMB-51 and KineticsSounds (those details are presented in Section 4 of the Supplementary). For compression corruption we use "glitchart" library whereas motion blur is implemented using 2D Filter using OpenCV applied to consecutive frames with changing angles and blur size to reflect motion. All three attacks are applied at random locations in the video to guarantee generalizability of the backdoor. We will refine the description of the experimental settings in the manuscript to ensure clarity and reproducibility. Code will also be released detailing the exact implementations of all attacks.
>
> **Methods Selected for Defense Evaluation**
> > A subset of the methods was selected for investigating backdoor defenses on UCF-101 dataset due to computational constraints and the desire to evaluate the most diverse and promising subset of attacks. Particularly, we showed BadNet (a visible patch based attack), FTrojan (an invisible frequency based attack), SIG (a more sophisticated variant of Blend), and two natural backdoor attacks which are visible yet natural (Frame Lag and Motion Blur). Given that the evaluated defenses either fail completely or can at best partially defend against existing attacks, the defenses are deemed useless regardless whether the other three attacks are evaluated or not. Nevertheless, in the revised manuscript we will include the results for the missing methods (video corruption, Blend, WaNet).
>
>
> **Novelty of Attacks:**
> >We emphasize that our utilization of motion blur, video corruption, and frame lag is novel within the field of video action recognition as backdoor triggers.  To the best of our knowledge, video corruption and frame lag have not been previously explored neither in context of adversarial attacks nor backdoor attacks. White motion blur has seen limited exploration, notably as adversarial attacks in references such as [1], it has not been utilized as a backdoor trigger. The essence of our claim rests on introducing these inherent video artifacts as backdoor triggers for the first time. In the final manuscript, we will cite [1] when discussing motion blur and better state that the novelty lies in applying those video artifacts as backdoor triggers in video action recognition.
>
> ### Minor Comments:
>
> **Selection Criteria:**
> > In the updated manuscript, we will explain why we chose certain methods, architectures, and datasets. Mainly, our choices are based on what's commonly used in video action recognition. We picked a range of attacks to study: from common patch attacks to subtle changes in pixel space all the way to frequency-based attacks.
>
> **Differing Attack Method Performance:**
> > UCF-101 typically yields higher clean data accuracy (CDA) and attack success rate (ASR) compared to HMDB-51. This is because UCF-101 is generally viewed as a less challenging dataset for video action recognition. HMDB-51, on the other hand, includes more nuanced actions, like distinguishing between "chewing" and "eating" or "laughing" and "smiling". Additionally, HMDB-51 contains some videos of lower quality. The variation in CDA reflects the inherent difficulty of the action recognition task on each dataset (see the baseline row versus attacked rows in Tables 1 and 2; the attacked CDA aligns closely with the baseline). Meanwhile, the ASR indicates the effectiveness of the attack, which is consistently high in both datasets, so any variations there are relatively minor.
>
> **Workflow Figure:**
> >A workflow figure is indeed a valuable addition to the paper, helping readers understand the experimental design. We will design and include an illustrative figure to present the overall pipeline of experiments.
>
> **Supplementary Material:**
> > Thank you for your positive comments on the supplementary material. Based on your suggestion, we will enrich the supplementary material with further implementation details and pertinent illustrations, similar to Figure 1. We are also going to release the code which will contain all implementation details of every attack and defense method studied in the paper.
>
> #### References
> >[1] Guo, Q., Juefei-Xu, F., Xie, X., Ma, L., Wang, J., Feng, W., & Liu, Y. (2020). ABBA: Saliency-Regularized Motion-Based Adversarial Blur Attack. ArXiv, abs/2002.03500.

---

### Review · Reviewer_cbm2 · 2023-10-27

**Summary Of Contributions:**

This paper revisits the traditional backdoor attack threat model and incorporate video-related aspects, such as video subsampling and spatial cropping.

It investigates audiovisual backdoor attacks against video action recognition models.

**Audience:**

Yes

**Broader Impact Concerns:**

Broader Impact Concerns are not addressed in the paper. As the paper discusses backdoor attack for video action recognition, it is better to discuss the ethical implications of the work, or potential negative effects.

**Claims And Evidence:**

Yes

**Requested Changes:**

see the weakness.

**Strengths And Weaknesses:**

\+ Through extensive experiments, it provides evidence that the previous perception of image backdoor attacks in the video domain is far from being comprehensive, especially when viewed in the poisoned-label attack setup.

\- The technical contribution of the paper may be limited. It mainly follows the basic backdoor attack framework to implement backdoor attack. For the video action recognition, although it considers some video-related aspects, such as dynamic attacks, frame lag, video compression, and sub-sampling, the ideas are straightforward and there are not many novelties. It also test the attack performance under different defenses. The defense methods mainly follow the previous works without new techniques. It also mentions the Audiovisual Backdoor Attacks. But the attack also follows traditional backdoor attack framework. Though the paper mentions many aspects, there are not many new techniques involved. The novelty may be limited.

\- It mentions clean label attack, but as the paper mainly focus on poisoned label attack, which is significantly different from clean label attack, it is not strange that the observation is different from the clean label attack.

\- Recent advances in transformer-based action recognition models have shown to achieve better performance on large training datasets compared to CNN-based models, such as (Arnab et al., 2021; Fan et al., 2021; Liu et al., 2022b; Bertasius et al., 2021). This paper mainly test the CNN-based models including I3D, TSM and SlowFast. The performance on the transformer-based models are still unknown. Since people may use   transformer-based action recognition models due to their better performance, it is better to have some results on transformer based models.

\- There are some related works, such as [R1,R2], which also investigates the backdoor attack for action recognition. It is better to discuss the difference and compare with them if possible.

\- I feel that the techniques in the paper mainly follow previous works, and the observations or conclusions are also similar to that in previous works. The attack, models and defenses in the experiments are not very new, mostly published before 2020. There are not something novel when I went through the paper.


[R1] Temporal-Distributed Backdoor Attack Against Video Based Action Recognition

[R2] Palette: Physically-Realizable Backdoor Attacks Against Video Recognition Models

---

> ### Author Response · Authors · 2023-11-06
> **Response to requested changes.**
>
> We thank the Reviewer for the comprehensive feedback and for highlighting the extensive experimental work conducted in our study. We appreciate the recognition of our efforts to delve into the nuanced domain of backdoor attacks in video action recognition. We have addressed each of your comments below, which we believe further highlight the contributions of our work:
>
> ### Weaknesses:
>
> **Technical Contributions and Novelty:**
> > While it's true that our paper builds on the foundation of traditional image backdoor attack frameworks, we believe that our work makes significant contributions by adapting and extending these frameworks to the relatively unexplored domain of video action recognition. The incorporation of video-related aspects such as dynamic attacks, frame lag, video compression, and sub-sampling may seem straightforward, yet they represent an essential step in understanding how backdoor attacks manifest in video data, which is inherently more complex than image data. Additionally, we highlight that existing image backdoor defenses fail against video backdoor attacks, shedding light on the importance of researching backdoor defenses tailored for the video domain. Our exploration of audiovisual backdoor attacks also extends the threat model to multimodal domains. We explore some concepts related to early-fusion and late-fusion of the multi-modal attacked video and audio networks which is previously unexplored revealing some interesting findings.
>
> **Poisoned vs. Clean Label Attacks:**
> > The focus on poisoned-label attacks was a deliberate choice due to their practical relevance and the lack of comprehensive studies on this setup in video domain. While clean-label attacks are indeed different, our observations shed light on the potential risks of poisoned-label setup. We will clarify our reasoning for this focus in the revised paper.
>
> **Transformer-based Models:**
> > Our goal in this work is to set foundations for video backdoor attacks, for that we focused on CNN based architectures which are widely adopted in video action recognition.
>
> **Related Works [R1,R2]:**
> > We thank you for pointing out these related works. Our paper aims to contribute to the ongoing conversation in the field of backdoor attacks on action recognition, and we will make sure to include a discussion of these papers in our revised manuscript. We will articulate the differences and similarties between our work and [R1, R2].
>
> **Novelty and Recent Techniques:**
> > In our paper, we studied a wide variety of approaches that belong to various categories such as patch-based attacks, warping-based attacks, and frequency-based attacks, covering most classes of backdoor trigger generation approaches. Existing backdoor defenses in the image domain are generally not easily extendable to the video domain, mostly due to computational restrictions (those limitations are discussed shortly in paragraph Defenses Against Video Backdoor Attacks page 8).
>
> Additionally, we will discuss the broader impacts of this work in a separate section in the paper.
>
> Again, we thank the reviewer for his time and efforts in reviewing our paper and for providing invaluable feedback to strengthen our paper.

---

### Decision · Action_Editor_8Bxo · 2023-11-28

**Recommendation:** Reject

**Comment:**

The key contribution this manuscript claims is to demonstrate the vulnerability of visual action recognition models to conventional backdoor attacks.    While Reviewer LLjH has turned a bit more positive about the manuscript (as the authors address their concerns), Reviewer cbm2 and GaPk provided a valuable point regarding the technical novelty. I put more weight on their recommendation.

My impression is that this paper may serve as a prior work on backdooring attacks against visual action recognition models, but even without this paper, the community already knows extending backdoor attacks to visual action recognition models is fairly straightforward. Reviewer cbm2 also mentioned some prior works exploring the same idea.

I therefore hesitate to say we should accept this paper.

**Audience:**

I think the audience of this paper could be someone working on backdoor attacks and defenses. As there are a few existing works on backdooring visual action recognition models, the paper could attract some of them. But as the results do not seem to be surprising (by the reviewers' comments), the manuscript may not offer insights different from what the community knows about backdoor attacks.

**Claims And Evidence:**

The paper shows that similar to conventional neural networks, those used for video action recognition are vulnerable to backdoor attacks. To demonstrate, the paper extends existing backdoor attacks to video recognition domains and offers empirical evaluations (also against existing defenses).

**Resubmission Of Major Revision:**

The authors may consider submitting a major revision at a later time.